# Drought and Elevated Carbon Dioxide Impact the Morphophysiological Profile of Basil (*Ocimum basilicum* L.)

T. Casey Barickman [1,*], Bikash Adhikari [1], Akanksha Sehgal [2], C. Hunt Walne [2], K. Raja Reddy [2] and Wei Gao [3]

1   North Mississippi Research and Extension Center, Mississippi State University, Verona, MS 38879, USA; ba917@msstate.edu
2   Department of Plant and Soil Sciences, Mississippi State University, Mississippi State, MS 39762, USA; as5002@msstate.edu (A.S.); chw148@msstate.edu (C.H.W.); krreddy@pss.msstate.edu (K.R.R.)
3   USDA UVB Monitoring and Research Program, Natural Resource Ecology Laboratory, Department of Ecosystem Science and Sustainability, Colorado State University, Fort Collins, CO 80523, USA; wei.gao@colostate.edu
*   Correspondence: t.c.barickman@msstate.edu; Tel.: +1-(662)-566-2201

**Abstract:** Treating plants with elevated carbon dioxide (eCO$_2$) can increase their drought tolerance. Increased atmospheric CO$_2$, a fundamental factor in climate change, may compensate for the drought-induced reduction in crop growth and yield. Basil, being moderately sensitive to drought stress (DS), experiences several morphological changes under DS. Thus, we designed an experiment that addresses how DS and different levels of CO$_2$ affect the overall morphological growth patterns during basil's early and late-season growth. The experiment was conducted under four different growth conditions: two water treatments, (1) a full-strength Hoagland's solution was added to the basil plants at 120% of the evapotranspiration each day, and (2) 50% of the full-strength Hoagland's solution was added to basil plants for the drought treatment, alongside two levels of CO$_2$ application [ambient 420 ppm (aCO$_2$) and elevated 720 ppm (eCO$_2$)]). The DS had a severe impact on the morphological traits of the shoot and root systems. Compared to control, DS reduced the marketable fresh mass (FM) by 31.6% and 55.2% in the early and late stages of growth. FM was highest under control + eCO$_2$ (94.4–613.7 g) and lowest under DS + aCO$_2$ (67.9–275.5 g). Plant height under DS + aCO$_2$ and DS + eCO$_2$ reduced by 16.8% and 10.6% during the late season. On the other hand, dry mass percent (DM%) increased by 31.6% and 55.2% under DS + eCO$_2$ compared to control in the early and late stages of growth, respectively. This study suggested that eCO$_2$ during DS significantly impacts basil morphological traits compared to aCO$_2$. Besides, anthocyanin decreased by 10% in DS + aCO$_2$ and increased by 12.6% in DS + aCO$_2$ compared to control. Similarly, nitrogen balance index, a ratio of chlorophyll and flavonoids, was recorded to be the highest in DS + aCO$_2$ (40.8) compared to any other treatments. Overall, this study indicates that the suppression of basil's morphophysiological traits by DS is more prominent in its later growth stage than in the earlier stages, and eCO$_2$ played an important role in alleviating the negative effect of DS by increasing the DM% by 55%.

**Keywords:** climate change; physiology; ambient carbon dioxide; root image analysis; chlorophyll; flavonoids; nitrogen balance index

## 1. Introduction

The progression of climate change has made the global agricultural system vulnerable and has negatively impacted overall agrarian production [1]. The increasing linear warming trend of 0.74 °C over 100 years (1906–2005) and an expected increment of 1.1 to 6.4 °C by the end of the 21st century have become a massive threat to agriculture [2,3]. Likewise, the increased atmospheric carbon dioxide (CO$_2$) is a fundamental factor in climate change and may compensate for the environmentally induced reduction in crop growth and yield [4]. The increasing level of global atmospheric CO$_2$ (increased by 40% in 2011, i.e.,

278–390.5 ppm, and 413.35 ppm in 2020) and its projected rise to 970 ppm by the 21st century suggests a disturbance in climatic resilience to these stress factors [3]. The difference in rainfall patterns comprises a decline in rainwater and increased rainfall intensity over a short period [5]. Thus, it is unequivocal that higher temperatures and uncertain rainfall patterns will increase drying conditions worldwide [1]. Moreover, the anticipated growing drought due to elevated atmospheric $CO_2$ and temperature will affect crops' growth and development, including basil (*Ocimum basilicum* L.) [6]. Therefore, it is crucial to assess climate change's influence on drought stress (DS) conditions for effective DS mitigation and crop adaptation [7,8].

Basil is a culinary and medicinal herb that grows best under warm climatic conditions [9], with an optimal temperature range of 25–30 °C [10]. However, a supplemental water supply is mandatory for the successful commercial production of basil. Several studies on the irrigation of basil have reported sensitivity to DS [11,12]. In general, basil ($C_3$ plants) is grown under a wide range of environmental conditions. The plant reacts to DS via a series of morphological and cellular responses [13]. Previous research has indicated that DS decreases the number and branching of the stems, decreases internode length size, and reduces plant height (Ht), leaf area (LA), nodal length, stem diameter, root, and shoot yield in basil [14,15].

Basil's response to elevated $CO_2$ (e$CO_2$) has not been appropriately explored in the past. However, e$CO_2$ is considered an innovative approach to improve plants' vegetative growth and nutritional value [16]. For example, e$CO_2$ increased the yield of leafy vegetables by 38% and stem vegetables by 17% [17]. e$CO_2$ also improves the biomass production in basil by 48% (Al Jaouni et al., 2018) and enhances chlorophyll content in the leaves [18]. A study on several leafy and stem vegetables reported that e$CO_2$ enhanced the total biomass, yield, and dry matter content [17]. These morphological and nutritional benefits in basil due to e$CO_2$ could help to increase its production efficiencies and nutritional value.

Many multidisciplinary approaches have reported the role and effect of DS and e$CO_2$ levels on crops under global climate change scenarios in recent years. However, very few studies have investigated the interactive effects of DS and e$CO_2$ on growth and basil development. Our understanding of the underlying implication of e$CO_2$ under DS conditions in basil is still inconclusive. A previous study reported that basil production increased up to 80%, increasing $CO_2$ levels from 360 to 620 ppm [4]. Similarly, the potential of DS tolerance in basil makes it an excellent alternative crop in dry regions [15,19] due to its high economic value. Besides, the study of basil's morphophysiological parameters under DS will help identify basil traits resistant to DS [20]. Thus, the current study's primary purpose is to understand the effect of DS coupled with e$CO_2$ on morpho-physiological attributes in basil.

## 2. Materials and Methods

### 2.1. Plant Materials and Growing Condition

Basil' Genovese (Johnny's Selected Seeds, Winslow, ME) seeds were sown in polyvinyl-chloride pots (15.2 cm diameter by 30.5 cm height) filled with a soil medium consisting of 3:1 sand/soil classified as a sandy loam (87% sand, 2% clay, and 11% silt) with a 500 g of gravel at the bottom of each pot. Six seeds were sown in each pot, and the plants were thinned to one plant per pot approximately seven days after emergence. Pots were organized in a randomized complete block design within a three-by-two factorial arrangement with temperature and $CO_2$ treatments. A total of four Soil-Plant-Atmosphere-Research (SPAR) chambers represents two blocks (ten replications each). Each SPAR chamber consisted of 3 rows of pots (ten pots per row). All environmental growing conditions were kept the same throughout the experiment except for irrigation volumes and $CO_2$. More detailed information on the SPAR chamber was earlier described by Reddy et al. [21] and Wijewardana et al. [22].

Basil plants were irrigated three times per day using an automated computer-controlled drip system with full-strength Hoagland's nutrient solution [23]. Irrigation was provided

at 700, 1200, and 1700 h, based on evapotranspiration values. Evapotranspiration rates expressed on the ground area (L·d$^{-1}$) throughout the treatment period were measured in each SPAR unit as the rate at which the cooling coils removed the condensate at 900-s intervals [21,24,25]. They were obtained by measuring the mass of water in collection devices connected to a calibrated pressure transducer.

### 2.2. Treatments Application

Basil plants were randomly assigned to each chamber consisting of 30/22 (day/night), in combination with ambient (420 ppm) (aCO$_2$) or elevated (720 ppm) (eCO$_2$) carbon dioxide concentrations. The daytime temperatures were initiated at sunrise and nighttime temperatures 1 h after sunset. There were two water treatments, imposed at 14 days after sowing (DAS) for the experiments: (1) a full-strength Hoagland's solution [23] was added to the basil plants at 120% of the evapotranspiration each day, and (2) 50% of the full-strength Hoagland's solution was added to basil plants for the DS treatment.

### 2.3. Phenology and Growth

Basil plants from each treatment combination were harvested to obtain phenotype and growth data on early and late-stage growth effects of DS and CO$_2$ at 17 and 38 days after treatment (DAT). Basil phenotypic data of Ht, node number (NN), branch number (BN), fresh mass (FM) were measured. Dry mass (DM) of the leaf (LDM), stem (SDM), root (RDM), shoot (ShDM), and whole plant (TDM) were measured for each treatment combination. Root to Shoot Ratio (RS) was measured using the ratio of RDM and ShDM.

LA was measured using the LI-3100 leaf-area meter (Li-Cor Bioscience, Lincoln, NE). Using a weighing scale, plant component FM was extracted from all basil plants. The plant FM samples were then dried for two days at 75 °C in a forced-air oven to yield basil DM. The DM percent (DM%) was calculated using (Shoot DM/FM) × 100%.

### 2.4. Root Image Acquisition and Analysis

Roots were cut and separated from the stems and washed thoroughly. The total root length (TRL) was determined using a ruler. The cleaned individual root systems were floated in 5 mm of water in a 0.3- by 0.2-m Plexiglas tray. Roots were untangled and separated with a plastic paintbrush to minimize root overlap. The tray was placed on top of a specialized dual-scan optical scanner (Regent Instruments, Inc., Quebec, QC, Canada) linked to a computer. Gray-scale root images were acquired by setting the parameters to high accuracy (resolution 800 × 800 dpi). Acquired images were analyzed for the lateral root length (LRL), root surface area (RSA), average root diameter (RAD), root volume (RV), number of root tips (RT), root forks (RF), and root crossings (RC) using WinRHIZO Pro software (Regent Instruments).

### 2.5. Morpho-Physiological Measurements

Leaf chlorophyll content (chlorophyll), epidermal flavonoids, epidermal anthocyanin, and nitrogen balance index (NBI) were measured on the second uppermost recently fully expanded leaf, second from the top, under each of three temperature treatments with a Dualex® Scientific Polyphenols and Chlorophyll Meter (FORCE-A, Orsay, France) at 38 DAT.

### 2.6. Data Analysis

Statistical analysis of the data was performed using SAS (version 9.4; SAS Institute, Cary, NC, USA). Data were analyzed using the PROC GLIMMIX analysis of variance (ANOVA) followed by mean separation. The experimental design was a randomized complete block in a factorial arrangement with two water and two CO$_2$ treatments, three-block, and ten replications. The standard errors were based on the pooled error term from the ANOVA table. Duncan's multiple range test ($p \leq 0.05$) was used to differentiate treatment classifications when F values were significant for main effects. Model-based

values were reported rather than the unequal standard error from a data-based calculation because pooled errors reflect the statistical testing. Diagnostic tests were conducted to ensure that treatment variances were statistically equal before pooling.

## 3. Results and Discussion

### 3.1. Morphological Traits

It is widely understood that basil thrives well under a 70% soil water capacity [26]. Our result indicated that the DS (water supply at 50% or less than 50% soil water capacity) affects the Ht, LA, and FM, as well as several other basil's morphological traits. A report on basil demonstrated that TDM yield and Ht decreased by 31.6% and 26%, respectively, due to DS [15]. Forouzandeh et al. [15] also reported several other morphological parameters such as that FM, ShDM, and RDM decreased by 42.2–60.1% under the DS with 60% soil water capacity in basil. Like DS, eCO$_2$ is also considered an important environmental factor in affecting the economic yield of C$_3$ plants such as basil [4]. Since these two factors occur concurrently, it is important to study the individual and combined factors to investigate potential interaction among factors. Morphological responses to DS in basil and most agronomic and horticultural crops include slow growth rate, reduced LA and LN, and increased RV and RS [27]. At 17 DAT, the interactive effect ($p < 0.001$) between DS + eCO$_2$ as well as DS + aCO$_2$ was observed on the Ht (Table 1). There was a significant decrease in Ht by 9.6% ($p < 0.05$) under DS + aCO$_2$ compared to control on 17 DAT. However, there was no difference for Ht of DS + eCO$_2$ compared to the control on 17 DAT. The decrease in Ht under DS + aCO$_2$ and DS + eCO$_2$ was observed by 16.8% and 10.6%, respectively, on 38 DAT, compared to control. Previous research indicated that DS in commercial basil cultivars significantly reduced Ht [28]. However, the eCO$_2$ can increase the Ht of basil by 8.5%, as reported by Singh et al. [29]. Consequently, the observed reduction in Ht in this study may be due to the disturbance in the basil metabolic process leading to poor cell division and elongation [30].

**Table 1.** Dry mass percent (DM%), plant height (Ht), node number (NN), branch number (BN), and leaf area (LA) of basil plants grown without drought stress (Control) and with drought stress at two levels of CO$_2$ (420 and 720 ppm) after 17 days of treatment.

| Treatment | DM% [1,3] | Ht | NN | BN | LA |
|---|---|---|---|---|---|
| | | 420 ppm | | | |
| Control | 8.269 [b] | 36.56 [a] | 7.1 [a] | 15.33 [a] | 1223.60 [ab] |
| Drought | 10.101 [ab] | 33.05 [b] | 6.9 [a] | 13.56 [a] | 997.72 [c] |
| | | 720 ppm | | | |
| Control | 8.981 [b] | 36.61 [a] | 7.0 [a] | 15.33 [a] | 1321.09 [a] |
| Drought | 11.823 [a] | 35.44 [a] | 7.0 [a] | 14.22 [a] | 1070.67 [bc] |
| Treatment [2,4] | ** | *** | ns | * | *** |
| CO$_2$ | ns | * | ns | ns | ns |
| Treatment × CO$_2$ | ns | * | ns | ns | ns |

[1] Dry mass in percentage (%); Height in centimeters (cm); Node number and branch number on a per plant basis; Leaf area units in centimeters squared. [2] Mean separation within the column by Duncan's multiple range tests; ns, *, **, *** indicates non-significant or significant at $p \leq 0.05$, 0.01, and 0.001, respectively. [3] Values followed by the same letter are not significantly different. [4] SE-Standard error of the mean, DM% = 0.7, Ht = 0.6, NN = 0.1, BN = 0.8, and LA = 59.4.

A previous study reported that DS reduced DM% by 31.5% in basil [15,28]. On the other hand, Singh et al. [29] demonstrated that DM increases by 34.4% in basil treated with 800 ppm CO$_2$. Moreover, eCO$_2$ decreases stomatal conductance and increases photosynthetic rates, reducing transpiration and higher water use efficiency [31,32]. Also, it was reported that CO$_2$ use becomes more efficient under DS when there is more supply of CO$_2$ [33]. As reported earlier, an increase in water use efficiency under DS due to eCO$_2$ also increases DM [34]. In support, eCO$_2$ under DS is expected to start carbon fixation and prolong active growth by maintaining the soil water reserved for longer [35]. Likewise, in the current study, the DM% increased significantly by 31.6% and 55.2% under DS + eCO$_2$

compared to control on 17 DAT and 38 DAT, respectively (Tables 1 and 2). Thus, it can be suggested that $eCO_2$ ameliorated the adverse effect of DS on DM by increasing the water use efficiency in basil through more carbon assimilation, which leads to more biomass accumulation.

**Table 2.** Dry mass percent (DM%), height (Ht), node number (NN), branch number (BN), and leaf area (LA) of basil plants grown without drought stress (Control) and with drought stress at two levels of $CO_2$ (420 and 720 ppm) after 38 days of treatment.

| Treatment | DM% [1,3] | Ht | NN | BN | LA |
|---|---|---|---|---|---|
| | | 420 ppm | | | |
| Control | 11.733 [b] | 61.67 [a] | 10.0 [a] | 29.87 [a] | 6946.3 [a] |
| Drought | 16.265 [a] | 51.27 [b] | 9.8 [a] | 31.27 [a] | 3913.3 [b] |
| | | 720 ppm | | | |
| Control | 10.677 [b] | 60.93 [a] | 10.1 [a] | 29.67 [a] | 8078.9 [a] |
| Drought | 16.571 [a] | 54.47 [b] | 10.0 [a] | 29.67 [a] | 3978.7 [b] |
| Treatment [2,4] | *** | *** | ns | ns | *** |
| $CO_2$ | ns | ns | ns | ns | ns |
| Treatment $\times$ $CO_2$ | ns | ns | ns | ns | ns |

[1] Dry mass in percentage (%); Height in centimeters (cm); Node number and branch number on a per plant basis; Leaf area units in centimeters squared. [2] Mean separation within the column by Duncan's multiple range test; ns and *** indicates non-significant or significant at $p \leq 0.05$ and 0.001, respectively; [3] Values followed by the same letter are not significantly different. [4] SE-Standard error of the mean, DM% = 0.5, Ht = 1.7, NN = 0.1, BN = 1.1, and LA = 581.36.

The leaf is considered the most drought-sensitive part of the plant [31]. It is also responsible for reducing water loss and promoting water use efficiency during DS [36]. In the current study, LA decreased by 19% under both DS + $aCO_2$ and DS + $eCO_2$ treatment on 17 DAT compared to control. Similarly, LA decreased by 43.6% and 50.8% under both DS + $aCO_2$ and DS + $eCO_2$ treatment on 38 DAT compared to control. Previous research indicated that basil LA decreased under DS [37]. Similarly, an experiment conducted on sunflowers also demonstrated a significant reduction in LA under the DS [38].

Conversely, different fruits, vegetables, basil, and several other $C_3$ plants treated with $eCO_2$ have demonstrated an increase in LA [29]. In the present study, $eCO_2$ fails to amend the effect of DS on LA, further supported by soybean reports [39]. In plants subjected to DS, their cell weakens, leading to low water potential and low turgor pressure and, ultimately, reduced growth [28], thus demonstrating that these factors are behind LA's reduction in basil [40].

In the current study, basil FM was also significantly ($p < 0.01$) reduced under the DS + $aCO_2$ and DS + $eCO_2$ treatment at 17 DAT and 38 DAT, which is further supported by previous research on basil [41]. It is worth noting that FM was highest under control condition at $eCO_2$ level and was lowest under DS at both $CO_2$ levels on 17 and 38 DAT (Tables 3 and 4). It is important to note that the water retention under DS + $eCO_2$ in the later season was poor, as shown in Table 4. A study in cork oak by Vaz et al. [42] reported that the effect of $eCO_2$ can deteriorate under any stress in the long run, which can make a difference in leaf morphology.

**Table 3.** Fresh mass (FM), leaf dry mass (LDM), stem dry mass (SDM), root dry mass (RDM), shoot dry mass (ShDM), total dry mass (TDM), and root-to-shoot ratio (RS) of basil plants grown under without drought stress (Control) and with drought stress at two levels of $CO_2$ (420 and 720 ppm) after 17 days of treatment.

| Treatment | FM [1,4] | LDM | SDM | RDM | ShDM | TDM | RS [2] |
|---|---|---|---|---|---|---|---|
| | | | 420 ppm | | | | |
| Control | 80.62 [b] | 4.479 [bc] | 2.188 [bc] | 0.941 [a] | 6.667 [bc] | 7.608 [bc] | 0.140 [bc] |
| Drought | 59.32 [c] | 3.987 [c] | 2.021 [c] | 1.066 [a] | 6.008 [c] | 7.073 [c] | 0.176 [a] |
| | | | 720 ppm | | | | |
| Control | 94.37 [a] | 5.779 [a] | 2.789 [a] | 1.021 [a] | 8.568 [a] | 9.589 [a] | 0.119 [c] |
| Drought | 67.94 [c] | 5.074 [ab] | 2.642 [ab] | 1.180 [a] | 7.717 [ab] | 8.897 [ab] | 0.163 [ab] |
| Treatment [3,5] | *** | ns | ns | ns | ns | ns | ** |
| $CO_2$ | ** | ** | ** | ns | ** | ** | ns |
| Treatment × $CO_2$ | ns | ns | ns | ns | ns | ns | ns |

[1] Fresh weight, leaf dry weight, stem dry weight, root dry weight, shoot dry weight, and total dry weight units on a gram per plant basis. [2] RS- Root to Shoot Ratio (Root Dry Mass/Shoot Dry Mass) [3] Mean separation within the column by Duncan's multiple range test; ns, **, *** indicates non-significant or significant at $p \leq 0.05$, 0.01, and 0.001, respectively. [4] Values followed by the same letter are not significantly different. [5] SE-Standard error of the mean, FM = 3.9; LDM = 0.3; SDM = 0.2; RDM = 0.1; ShDM = 0.5; TDM = 0.5; RS ratio = 0.01.

**Table 4.** Fresh mass (FM), leaf dry mass (LDM), stem dry mass (SDM), root dry mass (RDM), shoot dry mass (ShDM), total dry mass (TDM), and root-to-shoot ratio (RS) of basil plants grown under without drought stress (Control) and with drought stress at two levels of $CO_2$ (420 and 720 ppm) after 38 days of treatment.

| Treatment | FM [1,4] | LDM | SDM | RDM | ShDM | TDM | RS [2] |
|---|---|---|---|---|---|---|---|
| | | | 420 ppm | | | | |
| Control | 486.33 [b] | 25.032 [a] | 33.049 [ab] | 6.840 [ab] | 58.081 [ab] | 64.922 [ab] | 0.116 [b] |
| Drought | 284.30 [c] | 17.591 [b] | 27.591 [b] | 5.343 [b] | 45.182 [b] | 50.525 [b] | 0.120 [b] |
| | | | 720 ppm | | | | |
| Control | 613.71 [a] | 28.393 [a] | 38.733 [a] | 8.511 [a] | 67.126 [a] | 75.637 [a] | 0.128 [ab] |
| Drought | 275.46 [c] | 17.060 [b] | 29.756 [b] | 6.388 [b] | 46.816 [b] | 53.204 [b] | 0.140 [a] |
| Treatment [3,5] | *** | *** | * | ** | ** | ** | ns |
| $CO_2$ | ns | ns | ns | * | ns | ns | * |
| Treatment × $CO_2$ | ns | ns | ns | ns | ns | ns | ns |

[1] Fresh weight, leaf dry weight, stem dry weight, root dry weight, shoot dry weight, and total dry weight units on a gram per plant basis. [2] RS- Root to Shoot Ratio (Root Dry Mass/Shoot Dry Mass) [3] Mean separation within the column by Duncan's multiple range test; ns, *, **, *** indicates non-significant or significant at $p \leq 0.05$, 0.01, and 0.001, respectively. [4] Values followed by the same letter are not significantly different. [5] SE-Standard error of the mean, FM = 38.2; LDM = 2.1; SDM = 3.0; RDM = 0.7; ShDM = 5.0; TDM = 5.7; RS ratio = 0.006.

Other the other hand, a previous study demonstrated an increase in FM in basil by 54.1% under $eCO_2$ (827 ppm) + non-drought conditions (control) [43]. O'Leary et al. [43] also reported that, although $eCO_2$ helps to mitigate the negative effect of DS on FM through improved water use efficiency, $eCO_2$ always performs better under water-sufficient conditions (>70% soil water capacity). For this reason, FM was recorded the lowest under DS at both $CO_2$ levels in this study.

TDM and yield were reduced by 34% under deficit irrigation in different basil cultivars [26]. However, in our study, there was no interaction between DS and $CO_2$ treatments when analyzing DM%, NN, BN, LDM, SDM, RDM, ShDM, and RS compared to the control treatment on 17 DAT (Tables 1 and 3). On 38 DAT, there was no interaction effect between DS and $CO_2$ treatments on any morphological parameters (Table 4).

The root system is responsible for absorbing water and nutrients from the soil, and it plays an essential role in the plant's response to DS [36]. Some $C_3$ and $C_4$ plants have the robust ability to increase root growth at the early stage of DS to absorb water from the deep soil [44]. This suggests that the density, length, volume, and mass of roots are directly associated with crop DS resistance [45,46]. The root tissues were measured at 17 DAT (Table 5). However, none of the parameters (LRL, TRL, RSA, RAD, RV, RT, RF, or RC showed the interactive effects of the DS and $CO_2$ treatments. A treatment effect

($p < 0.01$) was observed in LRL, where there was a significant decrease in LRL by 13.5% under DS + eCO$_2$ compared to control at 17 DAT. Interestingly, RV, RT, RF, and RC increased by 13–20% under DS + eCO$_2$ than control under aCO$_2$. A report on cucumber demonstrated that an increasing CO$_2$ level from 400 to 1200 ppm increases RV and RT by 6–8% [47]. In a review by Rogers et al. [48], approximately 150 studies concluded that 92% of root growth increased with eCO$_2$, further supporting the current study's result.

**Table 5.** The mean of lateral root length (LRL), total root length (TRL), root surface area (RSA), average root diameter (RAD), root volume (RV), root tips (RT), root forks (RF), and root crossings (RC) of basil plants grown under without drought stress (Control) and with drought stress at two levels of CO$_2$ (420 and 720 ppm) after 17 days of treatment.

| Treatment | LRL [1,3] | TRL | RSA | RAD | RV | RT | RF | RC |
|---|---|---|---|---|---|---|---|---|
| | | | | | 420 ppm | | | |
| Control | 45.1 [a] | 4572.9 [a] | 854.3 [a] | 0.597 [a] | 14.00 [b] | 10,052 [b] | 38,545 [b] | 2412.6 [b] |
| Drought | 43.0 [ab] | 4230.7 [a] | 729.5 [a] | 0.547 [a] | 13.73 [b] | 14,347 [a] | 44,146 [b] | 3255.8 [ab] |
| | | | | | 720 ppm | | | |
| Control | 46.7 [a] | 4159.1 [a] | 738.6 [a] | 0.560 [a] | 15.45 [ab] | 12,477 [ab] | 46,580 [ab] | 3287.8 [ab] |
| Drought | 40.4 [b] | 4265.6 [a] | 765.1 [a] | 0.574 [a] | 17.60 [a] | 15,042 [a] | 55,344 [a] | 3840.4 [a] |
| Treatment [2,4] | ** | ns | ns | ns | ns | * | ns | ns |
| CO$_2$ | ns | ns | ns | ns | * | ns | * | * |
| Treatment × CO$_2$ | ns | ns | ns | ns | ns | ns | ns | ns |

[1] Lateral root length, total root length, and root average diameter on a centimeter per plant basis; root surface area, root volume on a cubic centimeter basis; root tips, root forks, and root crossings on a number per plant basis. [2] Mean separation within the column by Duncan's multiple range test; ns, *, ** indicates non-significant or significant at $p \leq 0.05$, 0.01, and 0.001, respectively. [3] Values followed by the same letter are not significantly different. [4] SE-Standard error of the mean, LRL = 1.5; TRL = 258.3; RSA = 52.9; RAD = 0.02; RV = 1.3; RT = 1723.8; RF = 4462.6; RC = 370.2.

### 3.2. Physiological Measurements

Drought is a significant factor for damaging the photosynthetic pigments and thylakoid membranes [49]. DS also inhibits plants' photosynthetic apparatuses by declining CO$_2$ availability and stomatal closure [50]. To study basil's leaf physiology changes under DS, different physiological parameters such as chlorophyll content, flavonoids, anthocyanin, and NBI were measured (Table 6). Flavonoid is a ubiquitous secondary metabolite in plants, which helps to protect the plant from abiotic and biotic stresses, while anthocyanin reduces the damage caused by free radical activity [51]. Both anthocyanin and flavonoid compounds are responsible for antioxidant activity in plants [52]. Both compounds increased under the DS + eCO$_2$ conditions [53,54]. However, in the present findings, the flavonoid was indifferent to the control treatment under the DS + eCO$_2$ condition, which contradicts the earlier report on basil by Al Jaouni et al. [4]. Previous research demonstrated that anthocyanin decreased under DS + aCO$_2$ but increased under DS + eCO$_2$ [53,55]. These reports support the recent finding where anthocyanin decreased by 10% in DS + aCO$_2$ and increased by 12.6% in DS + eCO$_2$ compared to control. Similarly, NBI, a ratio of chlorophyll and flavonoid, was measured, and it was recorded to be the highest in DS + aCO$_2$ (40.8) compared to any other treatments. A study by Taub and Wang [56] reported that plants grown under eCO$_2$ had decreased nitrogen concentration compared to plants grown under aCO$_2$.

**Table 6.** The mean of leaf chlorophyll, flavonoid, anthocyanin, and nitrogen balance index (NBI) of basil plants grown without drought stress (control) and with drought stress at two levels of $CO_2$ (420 and 720 ppm) after 17 days of treatment.

| Treatment | Chlorophyll [3] | Flavonoids | Anthocyanin | NBI [1] |
|---|---|---|---|---|
| | [$\mu g \cdot mL^{-1}$] | [$mg \cdot g^{-1}$ DM] | [$mg \cdot g^{-1}$ DM] | |
| | | 420 ppm | | |
| Control | 21.468 [bc] | 0.6853 [ab] | 0.1144 [b] | 32.415 [b] |
| Drought | 25.744 [a] | 0.6455 [b] | 0.1028 [c] | 40.890 [a] |
| | | 720 ppm | | |
| Control | 18.978 [c] | 0.7044 [ab] | 0.1126 [bc] | 28.062 [c] |
| Drought | 22.027 [b] | 0.7394 [a] | 0.1269 [a] | 30.391 [bc] |
| Treatment [2,4] | *** | ns | *** | *** |
| $CO_2$ | ** | * | ** | *** |
| Treatment $\times$ $CO_2$ | ns | ns | ns | * |

[1] NBI-Nitrogen Balance Index (a ratio of chlorophyll and flavonoid). [2] Mean separation within the column by Duncan's multiple range test; ns, *, **, *** indicate non-significant or significant at $p \leq 0.05$, 0.01, 0.001, respectively. [3] Values followed by the same letter are not significantly different. [4] SE-Standard error of the mean, Chlorophyll = 0.9; Flavonoid = 0.03; Anthocyanin = 0.04; NBI = 1.600.

Similarly, DS is also responsible for decreasing the nitrogen isotope composition and the transient decrease in chlorophyll, which increases the accumulation of anthocyanin [57]. In the present study, chlorophyll increased by 20% and 16% under DS when $aCO_2$ and $eCO_2$ were applied, respectively, compared to control. In brief, DS + $eCO_2$ promotes chlorophyll and inhibits NBI, increasing the accumulation of anthocyanin.

## 4. Conclusions

This study provides evidence that DS + $eCO_2$ has a significant positive impact on basil's overall morphology. $eCO_2$ remarkably reduced the negative effect of DS by promoting several morphological traits such as DM, RV, RT, RF, and RC. The DS had a severe impact on several morphological traits comprising both shoot and root systems. Compared to control, the DS reduces the marketable FM remarkably by 31.6% and 55.2% in the early and late basil season. FM is the highest under control + $eCO_2$ (94.4–613.7 g), while it was the lowest under DS + $aCO_2$ (67.9–275.5 g). Similarly, Ht reduction under DS + $eCO_2$ (10.6%) is significantly lower than DS + $aCO_2$ (16.8%) during the late season. DM increases by 31.6% and 55.2% under DS + $eCO_2$ compared to control in the early and late season, respectively.

This study suggests that $eCO_2$ during DS has a more significant positive effect on basil morphological traits than $aCO_2$. Also, $eCO_2$ positively impacted and increased the NBI and chlorophyll by alleviating the negative impact of DS. Conversely, $eCO_2$ failed to lessen the adverse effect of DS on FM, LA, and Ht. Overall, this study indicates that DS impacted the basil more strongly in the late rather than in the early season, and $eCO_2$ in the late season has a more significant impact on some basil's morphological traits such as LA, FM, RDM, ShDM, and TDM than $aCO_2$.

**Author Contributions:** T.C.B.: conceptualization, methodology, validation, formal analysis, investigation, resources, data curation, writing—original draft, writing—review & editing, visualization, supervision, project administration, funding acquisition. B.A.: formal analysis, writing—original draft, writing—review & editing. A.S.: methodology, validation, investigation. C.H.W.: methodology, validation, formal analysis, investigation. K.R.R.: conceptualization, methodology, validation, formal analysis, investigation, resources, data curation, writing—review & editing, visualization, supervision, project administration, funding acquisition. W.G.: conceptualization, methodology, validation, resources, funding acquisition. All authors have read and agreed to the published version of the manuscript.

**Funding:** This material is based on the work supported by the USDA-NIFA Hatch Project under accession number 149210, and the National Institute of Food and Agriculture, 2019-34263-30552, and MIS 043050 funded this research.

**Institutional Review Board Statement:** Not applicable.

**Informed Consent Statement:** Not applicable.

**Data Availability Statement:** The data presented in this study are available on request from the corresponding author.

**Acknowledgments:** We thank David Brand for technical assistance and graduate students at the Environmental Plant Physiology Laboratory for their help during data collection. We would also like to thank Thomas Horgan for his technical assistance on the project.

**Conflicts of Interest:** The authors declare that they have no known competing financial interests or personal relationships that could have appeared to influence the work reported in this paper.

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
