# Peer review of "Drought and Elevated Carbon Dioxide Impact the Morphophysiological Profile of Basil (Ocimum basilicum L.)"

_2673-7655, doi:10.3390/crops1030012_

Round 1

Reviewer 1 Report

Dear authors,

the manuscript is very well written, however in the title and in the abstract, there is no single word about the physiological results (pigments, nitrogen etc)

Please include these results in the abstract. 

Also, the sentence in the abstract should be rewritten LINE no 21-22: Height reduction under drought + eCO2 (10.6 %) was significantly lower than drought + aCO2

Hight reduction was significantly lower... please try to express it in other words.

Best wishes

(16.8 %) during the late season.  

Reviewer 2 Report

The review on the publication by Barickman et al. under the title Drought and Elevated Carbon-dioxide Impact the Morphological Profile of Basil (Ocimum basilicum L.)

In general, the publication is well prepared, and I have only minor comments to it.

Line 33-34 Change from 0C to 0C

I think authors have to include some figures of the basil in the publication as in the title; we can see the following words - the changes in morphological profile. Can you see any changes by the naked eye?

Reviewer 3 Report

Albeit there are some interesting points and data could merit publication, the way data is presented is very weak and difficults the understanding and evaluation fo the results. Please mind the following points.

a) English use and grammar is very weak. Consider a professional edition.

b) The term morphology, in the way is used here, is very confussing or even I would say, incorrect. Morphology refers to the plant shape, but what authors are studying is yield and stress resistance. Please, correct it.

c) How is DM % calculated. Is not explained in mat. and met. Is referred to fresh weight? Please explain. Is difficult to understand how this value increases upon drought stress.

d) Each value in tables should have its own standard error, in order to evaluate the accuracy.  Presenting a mean of the errors is confusing.

Reviewer 4 Report

Authors attempted to describe their investigation of the effect of elevated CO2 on basil plants grown in drought conditions – to find interactive effects. Unfortunately, in my view the data in the manuscript are presented in such a chaotic way that the aim was not achieved. The effect is opposite, the paper scream to be rejected. However, I believe that some data there might be publishable, and it is up to authors to start over and make this paper worthy and resubmit. The main problem is that there are too many insignificant results put into flow with important one and all of it is mixed up with some literature data. It is very very very hard to follow. I would recommend making clear result section and then discussion. While doing that authors have a task to catch all the mistakes/contradictions in data interpretation that overflow the manuscript. Authors must find a way to present data in readers friendly way! I tried to highlight some things, that I cannot get my mind around (in the attached pdf file) but it was so confusing I just got lost and tired and decided to leave it for authors. My advice, that authors may ignore, is to first choose the results that do differ significantly and put them into graphs (box plots are the best but I can live with bars) – those result should tell the main story about role of eCO2 in the counteracting drought effects. There is no need to write results how they were gathered or in any particular order other then one that answer scientific question. Next authors might choose most surprising results that did not change (few) and graph them, debate what is going on. Finally, authors can refer to one big table of no significantly changed results and comment on meaning of that. Additionally, the aCO2 is a control to eCO2 so ambient CO2 should not be referred as a treatment. In conclusions I would advise to focus on eCO2+drought, and avoid making obvious statements about drought itself, as it is a subject that was studied in plants starting from Aristoteles, Theophrastus et al. 350-287BCE. Good luck.

Round 2

Reviewer 1 Report

The manuscript is acceptable in its present form.

Author Response

Please see the attached response to reviewers. Thank you!

Reviewer 3 Report

Paper can be accepted

Author Response

(The authors gave the same response as above.)

Reviewer 4 Report

I believe authors didn’t got an attached PDF file with my highlights and comments (attached again). As I mentioned in “my paragraph” there is so many mistakes that I will not attempt to correct all of them – it is not my job to write paper for authors. Just to give an example what need attention:

First sentence of the abstract:

To ameliorate the drought effect, exposing plants to elevated CO2 can significantly increase tolerance

  1. There are two parts with no connection (wrong syntax) It might be that authors would like to say “Growing plants in elevated CO2 can increase drought tolerance”
  2. I also think that for this premature statement – why drought and CO2 levels must be synergistic and not opportunistic? – This is rather the aim of this paper, to explain that relationship in basil…

The second sentence:

“Thus, this experiment addresses how drought stress and different levels of CO2 affect the morphology during basil’s early and late-season development.”

  1. This should be a manuscript not lab report about experiments
  2. As other reviewer mentioned the results are not about basil morphology only… so authors should correct ALL part of manuscript.
  3. Remove development - it is confusion to what it refers to: seasons or basil. Could be growth.
  4. If the relationship between CO2 and drought is known (1st sentence), the paper about what are effects of elevated CO2 impact on plant in drought is not novel… so those two sentences create chaos for reader – why this research is important?

The third sentence:

“The experiment was assigned under two growth conditions (control and drought stress) followed by two levels of CO2 application [ambient 420 ppm (aCO2) and elevated 720 ppm 17 (eCO2)].”

  1. “assigned” is wrong word – in this context: the experiments were performed under drought and eCO2 conditions (much simpler to read)
  2. Ambient 420 ppm CO2 level is also a control to eCO2, like non-drought conditions are control to drought – there is a lot of knowledge about drought in ambient CO2 and authors do not show anything significant there. If authors mean constant (controlled CO2 level) that is a different story – but based on the manuscript it is not the case.
  3. Word “followed” might suggest that first was drought and then CO2 treatment – but in methods the treatments were described as concurrent. That completely change the context of the paper.
  4. Authors could write that: We tested the effect of increased CO2 on drought tolerance in basil plants.

Next authors literally list some of their results and many times miss verbs or use wrong syntax. I mean it is an abstract – the result should be mentioned but only the most significant ones, not a whole list…

Line 30 is an example of lack of logic:

 “Also, there was a positive impact of aCO2 on the physiological traits under DS.”

How ambient CO2 can have positive impact on any traits – like this is a reference point – a control to compare eCO2. So, it has positive impact compared to what? Authors have to explain what they mean.

Last sentence (missing the point of paper):

“Overall, this study indicates that drought-impacted basil’s morphophysiological traits later in its growth stage more than the earlier stages of growth.”

  1. This study is about the effects of eCO2 and drought, so why the overall conclusion refers only to the drought impact.
  2. The syntax is wrong – it is not clear what is impacted when

The rest of the manuscript is like the abstract, it is full of contradictions (illogical statements, wrong conclusions etc.) and syntax errors (English sentence construction) that make this paper unpublishable at this stage.

Just for sake of example (out of many), first part about DM% and the line 184-186 “It can be suggested that the adverse effect of drought stress on DM was ameliorated by eCO2 in basil through the sufficient carbon assimilation that leads to more biomass accumulation.”

  1. What sufficient mean – like is it more, less? In principle this sentence just say that elevated CO2 led to more CO2 that plant could uptake – which is obvious.
  2. More biomass is from assimilating more CO2, but that is also true for no drought control (which just show substrate shortages for photosynthesis in both control and DS). Authors should relate the drought issue – e.g. focus on water use that might be more efficient while CO2 is elevated – e.g. shorter time when stomata have to be open or smaller stomata aperture… etc…
  3. This DM result is just not put into a physiological context: e.g. eleveated CO2 in both control and DS also lead to reduced LCHL in control aCO2 to control eCO2 (12% decrease) and DS aCO2 to DS eCO2 (14% decrease). So there is less chlorophyll,  more biomass – so, maybe photosynthesis is more efficient in higher CO2…
  4. Even more, how authors explain relationships like fresh biomass and DM%(or TDM). As written FM (late season) was elevated by eCO2 in control but decreased by eCO2 in DS. That is not the case in early season – what is the reason? How authors propose, the CO2 interplay with water management in plants? – Why water retention is worse upon eCO2 later in the season? And how it is related to higher DM? Above could make the eCO2 efficient stomata management argument weaker – but maybe not since there is no late season chlorophyl data. If only authors could arrange this work into meaningful results that are worth debating… The list of ups and downs is useless without context and finding relationships with own research, not only confirmation and speculation by literature, the way the manuscript is writen make it very hard to find meaning.

In the PDF I attached before (an again now) I point some other problems, but authors in my opinion should rewrite the result and discussion part. If authors want just to publish (and do not care about impact of the paper), it should be enough to make:

  1. separate list of results with figures instead of tables,
  2. discussion section that highlights the most significant results (complex, relationship between relevant ones e.g. FM -TDM), literature context all of it regarding the impact of eCO2 on plant phenotype in drought. This should be enriched by attempt to hypothesize what given result mean in context of eCO2/drought and not only that something is up/down.

To summarize, the whole paper is written in readers hostile manner. Authors failed to arrange results in meaningful piece and fail to answer why certain traits are affected by eCO2 in plants grown in drought and others not. There is no attempt to point, not to mention explain the mechanisms of morphophysiological changes. I recommend authors to revised it as whole, including rewriting and simplifying results, finding relationship between results. Authors must add graphs to represent most significant results.

Finally, I review papers thoroughly and always present my own conclusions with as specific points as manuscript allows. This submission does not allow for me to get into more detail since, I am not used to writing papers for authors. Submitting the manuscript in kind of draft form is disrespectful for peer-reviewers.

Author Response

(The authors gave the same response as above.)
